# Phylogenetic conservatism drives nutrient dynamics of coral reef fishes

Jacob E. Allgeier [1✉], Brian C. Weeks[2], Katrina S. Munsterman[1], Nina Wale [3], Seth J. Wenger[4], Valeriano Parravicini [5,6], Nina M. D. Schiettekatte [5,6], Sébastien Villéger [7] & Deron E. Burkepile[8,9]

The relative importance of evolutionary history and ecology for traits that drive ecosystem processes is poorly understood. Consumers are essential drivers of nutrient cycling on coral reefs, and thus ecosystem productivity. We use nine consumer "chemical traits" associated with nutrient cycling, collected from 1,572 individual coral reef fishes (178 species spanning 41 families) in two biogeographic regions, the Caribbean and Polynesia, to quantify the relative importance of phylogenetic history and ecological context as drivers of chemical trait variation on coral reefs. We find: (1) phylogenetic relatedness is the best predictor of all chemical traits, substantially outweighing the importance of ecological factors thought to be key drivers of these traits, (2) phylogenetic conservatism in chemical traits is greater in the Caribbean than Polynesia, where our data suggests that ecological forces have a greater influence on chemical trait variation, and (3) differences in chemical traits between regions can be explained by differences in nutrient limitation associated with the geologic context of our study locations. Our study provides multiple lines of evidence that phylogeny is a critical determinant of contemporary nutrient dynamics on coral reefs. More broadly our findings highlight the utility of evolutionary history to improve prediction in ecosystem ecology.

[1] Department of Ecology and Evolutionary Biology, University of Michigan, Ann Arbor, MI, USA. [2] School for Environment and Sustainability, University of Michigan, Ann Arbor, MI, USA. [3] Department of Microbiology and Molecular Genetics and Department of Integrative Biology & Ecology Evolution and Behavior Program, Michigan State University, Lansing, MI, USA. [4] Odum School of Ecology, University of Georgia, Athens, GA, USA. [5] PSL Université Paris: EPHE-UPVD-CNRS, USR 3278 CRIOBE, Université de Perpignan, Perpignan, France. [6] Laboratoire d'Excellence "CORAIL", Perpignan, France. [7] MARBEC, Université de Montpellier, CNRS, IFREMER, IRD, Montpellier, France. [8] Department of Ecology, Evolution, and Marine Biology, University of California, Santa Barbara, CA, USA. [9] Marine Science Institute, University of California, Santa Barbara, CA, USA. ✉email: jeallg@umich.edu

In many ecosystems, consumer communities drive fundamental processes, e.g., secondary production, nutrient recycling, consumption[1–5], and the importance of consumer diversity for these processes is well established[6,7]. A central mechanism by which diversity can promote ecological processes is simply through the presence or absence of particular species that contribute disproportionately to a particular function[8,9]. For example, in a species-rich tropical river system a single species, *Prochilodus maria*, disproportionately influences consumer-mediated nutrient cycling because it stores substantial amounts of nutrients in its tissues and recycles large amounts of nutrients via excretion[10]. Thus, understanding which attributes of a given species best explain their relative contribution to ecosystem processes is imperative for improving predictions of consumer-mediated processes and the impacts of biodiversity loss on ecosystems.

Chemical traits that mediate ecological processes, here defined as nitrogen (N), phosphorus (P), or carbon (C) stored in body tissues, or N and P excretion rates, like any species-level trait, are the result of both ecological context[11] and evolutionary history[12–14]. At the scale of the individual, these "chemical traits" are maintained on ecological timescales by biological and chemical feedbacks that enable the organism to resist changes in chemical composition when faced with changes in internal (physiological) and external (environmental) conditions[15–17]. On short timescales, these feedbacks should be particularly pronounced for processes like excretion; e.g., an animal with low physiological demand for N (low body %N) feeding on N-rich foods would be expected to excrete N at high rates. However, on longer timescales, these feedbacks should also influence the life histories of consumers such that over evolutionary time they minimize the mismatch between dietary intake and nutritional demand[17]. For example, herbivores are expected to have lower body nutrient content because of their nutrient-poor diets, and thus they are predicted to have relatively low nutrient excretion rates because their food resource adequately meets their nutritional demand[15–17].

The evolutionary histories of species are also important in shaping ecologically relevant traits[8,13,18–20]. Evolutionary processes, such as divergent selection, can result in increased trait variation within- and among-species when populations become geographically isolated and are exposed to different ecological contexts[21]. Conversely, as taxonomic groups (e.g. families) accumulate species, the shared evolutionary and biogeographic histories can result in traits being similar among species within a clade[12,13]. For example, if body phosphorous (P) is a conserved trait, the loss of P-rich boney structures in the common ancestor of a clade would be expected to result in lower body P content in its descendant species. This might result in a phylogenetically conserved increase in excretion of P due to reduced physiological demand for P within the clade. Importantly, this phylogenetic conservatism is a mechanism by which contemporary trait variation can be generated through a dynamic biogeographical and evolutionary history, rather than simply in response to contemporary ecological interactions[14]. Community ecology has long embraced the need to integrate evolutionary history and ecology to understand variation in traits that determine community dynamics and assembly[12–14]. However, the extent to which ecology and evolution determine variation in traits relevant for ecosystem processes is poorly understood and represents an important challenge for ecosystem ecology[19,20,22,23].

Coral reefs are among the most productive ecosystems on the planet but paradoxically persist in highly nutrient-poor environments. One mechanism by which these high rates of productivity can be achieved is to have large amounts of stored nutrients within ecosystem biomass and tight cycling between the pools of nutrients[24,25], i.e., nutrient capacity[26]. Fishes are now recognized to be fundamental for coral reef nutrient capacity[27], storing large amounts of nutrients in their tissue and supplying bioavailable nutrients for primary producers at high rates[28,29]. Here we assess the relative importance of contemporary ecological factors and evolutionary history in driving variation in chemical traits, across two diverse radiations of coral reef fishes from the Caribbean and Polynesia that diverged at least 3 million years ago[30]. We measured nine chemical traits (excretion rates of N and P and their ratio—N:P, and body content of C, N, and P, and their ratios—C:N, C:P, N:P) on 1525 individuals from 178 fish species (805 individuals from 107 species, and 656 individuals from 71 species in Polynesia and the Caribbean, respectively; Supplementary Table 1) across 40 families (11 shared between regions) that span a large range in body mass (0.08–2597 g). Importantly, our study included fishes that persist in similar environmental and ecological conditions, i.e., temperature, habitat types, but the contrast in nutrient limitation because of underlying differences in specific study locations (N limitation in Mo'orea, French Polynesia[31,32] vs. P limitation in The Bahamas, Caribbean[33–35]). We analyzed these data within a comparative phylogenetic framework to understand the relative importance of ecological factors associated with nutrient acquisition (diet, body size, nutrient demand, etc.) and phylogenetic relatedness as drivers of trait variation. Specifically, we address three questions:

1. To what extent is trait variation explained by ecological factors or phylogenetic relatedness?
2. How does the relative strength of phylogenetic conservatism differ between the Caribbean and Polynesia?
3. To what extent does biogeographic history influence trait variation beyond that predicted by phylogenetic divergence?

We show that a fish species' evolutionary history is overwhelmingly the best predictor of how it contributes to nutrient cycling, outweighing the importance of ecological factors that are often thought to be key drivers. Variation in fish traits that contribute to nutrient cycling across regions was, however, consistent with biogeographic differences in nutrient limitation, highlighting that ecological factors may drive trait evolution in the long term. We demonstrate that to explain contemporary ecosystem dynamics we need to consider past evolutionary processes.

## Results

**Phylogenetic history best explains chemical trait variation across the Caribbean and Polynesia.** We used Bayesian hierarchical models that accounted for phylogenetic relatedness among species to compare the extent to which ecological factors associated with contemporary nutrient acquisition and phylogenetic history explain the variation in each trait across both regions (Methods). The ecological factors (included as fixed effects) were: trophic grouping (TG), natural abundance of stable isotopes ($\delta^{13}C$ and $\delta^{15}N$—indicating relative resource breadth, and position in the food chain, respectively; "Methods"), body mass, and body nutrient demand—indicating relative demand for important macronutrients (e.g., Body %N, P, C; "Methods";[19,22,36]). The region was also included as a fixed effect in all models.

Variation in all chemical traits was better explained by phylogeny than by ecological factors (Fig. 1), reflecting strong phylogenetic structure in the data. Specifically, the ecological factors only explain a small proportion of the variation in the data (parameters in Fig. 1B are represented by "*Ecology*" in Fig. 1A) relative to the full model that also accounts for phylogeny, and were largely not significant or had small effect sizes (Fig. 1B).

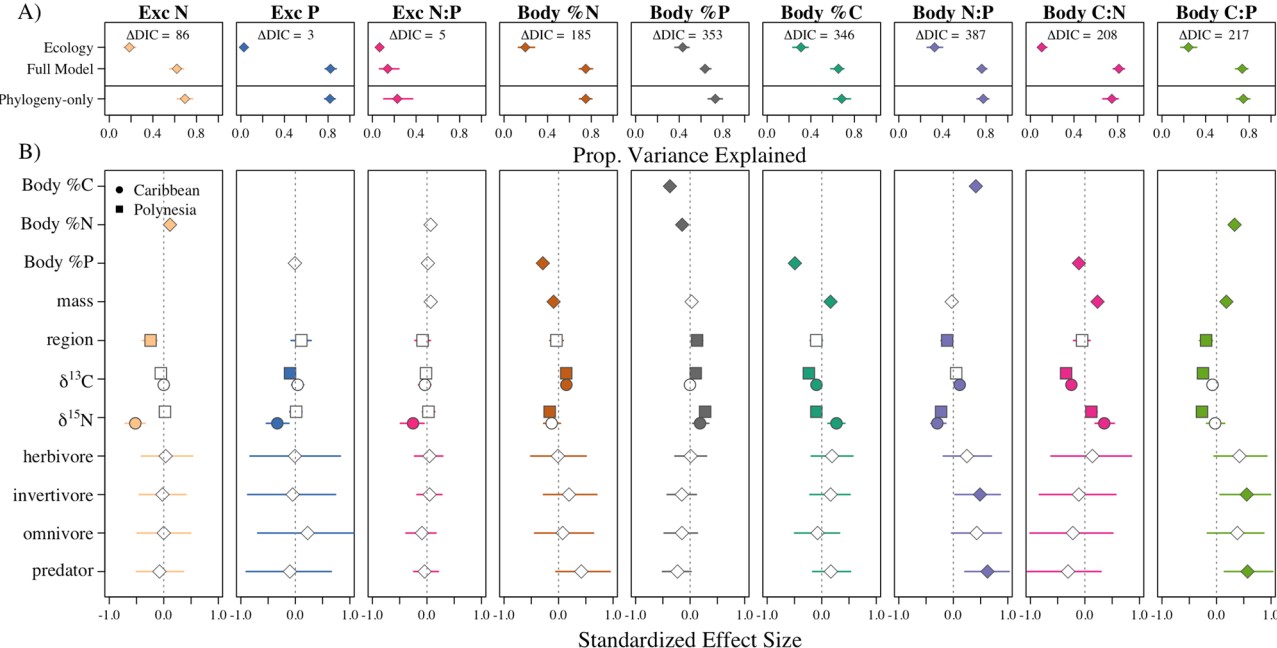

**Fig. 1 Phylogenetic relatedness is the best predictor of chemical trait variation in coral reef fishes. A** Variance explained (diamonds) by the "ecology" terms in the model (fixed effects), the "full model" (fixed plus random effects), and the "phylogeny-only" models for each chemical trait where: "Exc" indicated excretion for N, P, or their ratio N:P (molar), and "Body" indicates percent C, N, P, and their ratios N:P, C:N, C:P (molar) of dry mass ($n = 1266$, 1133, 1123, 1255, 1255, 1255, 1255, 1255, and 1255 individuals, respectively). Error bars indicate 95% Bayesian credible intervals (CIs) associated with model error. ΔDIC indicates the best model with a value >2 showing favor for the full model. **B** Standardized effect sizes and CIs (diamonds) of all ecology variables in the full model (all fixed effects). Error bars indicate 95% Bayesian credible intervals (CIs) associated with model error. Body nutrient predictors (e.g., Body %P) are not in all models due to strong covariance among the variables and were chosen as such to test specific hypotheses. Exc N and P are mass-corrected values and thus 'mass' is not tested in these models—see Supplementary Fig. 1 for models that are not mass-corrected; these models yield equivalent results. Circles indicate estimates for the Caribbean fishes. Squares indicate estimates for Polynesian fishes and in the case of 'region' indicate if the chemical trait significantly differs from the Caribbean (the model intercept). Filled points indicate the CIs do not overlap with zero.

Although the full model did have a lower deviance information criterion (DIC) in all cases and thus performed better than the phylogeny-only (intercept-only) model—an intuitive outcome because at least some of the ecological factors were significant in the models (Fig. 1B), the intercept only-model explained roughly the same amount of variation in the data as the full model (Fig. 1A). This highlights that, even though including the ecological terms is selected for by DIC, their inclusion does not improve the overall predictive power of this model, and thus phylogenetic relatedness alone is very good at explaining chemical trait variation (typically explaining 80% of the variation in the data; Fig. 1A). These results suggest that chemical traits are strongly phylogenetically conserved.

**The relative strength of phylogenetic conservatism differs between geographic regions.** All chemical traits were strongly phylogenetically conserved as indicated by the high Pagel's $\lambda$ (Fig. 2A), a measure of the phylogenetic signal, defined as the tendency for relatives to resemble each other more closely in their characteristics than expected by chance[12]. The degree of phylogenetic conservatism differed substantially across regions with higher levels of chemical trait conservatism in the Caribbean than in Polynesia (Fig. 2A and Supplementary Fig. 2; "Methods").

We hypothesized that traits with lower levels of phylogenetic conservatism would be better explained by contemporary ecological factors as they are less constrained by evolutionary history. In Polynesia, where the community is substantially more species-rich and phylogenetic trait conservatism is lower, we found three pieces of evidence to support this hypothesis: (1) greater resource breadth (wider standardized distribution of $\delta^{13}$C; Fig. 2Bi), (2) longer food chain length (wider standardized

distribution of $\delta^{15}$N; Fig. 2Bii), and (3) larger niche width, i.e., standardized measures of the standard ellipse area (SEA), and convex hull (Fig. 2Biii, "Methods"). Although field sampling effort was similar in both regions ("Methods"), to further account for potential sampling biases in our analyses, comparisons between regions were made by iteratively resampling a subset of individuals from each region 500 times (this iterative process was performed three times using 250, 375, or 450 individuals; "Methods"). The co-occurrence of weaker phylogenetic trait conservatism, increased species diversity, and increased niche diversification collectively are consistent with the idea that ecological interactions may play a more important role in driving chemical trait variation in Polynesia than the Caribbean.

**Across-region differences in chemical traits are consistent with biogeographical variation in resource limitation.** The differences in underlying geology in our study locations in The Bahamas and Mo'orea, French Polynesia offer a novel opportunity to examine if differences in long-term resource limitation can drive variation in chemical traits. Using phylogenetically corrected models we found that four of the nine traits were significantly different across regions (Fig. 3A). That is, across the two locations these four chemical traits differed beyond what would be expected based on the strong phylogenetic signal in our data—specifically, body P was higher, body N:P and C:P were lower, and N excretion was lower in Mo'orea than in The Bahamas. Importantly, these differences were consistent with expected differences in relative N and P availability in each study location: volcanic high (elevation)-islands such as Mo'orea should be more N limited[31,32], and carbonate low (elevation)-islands,

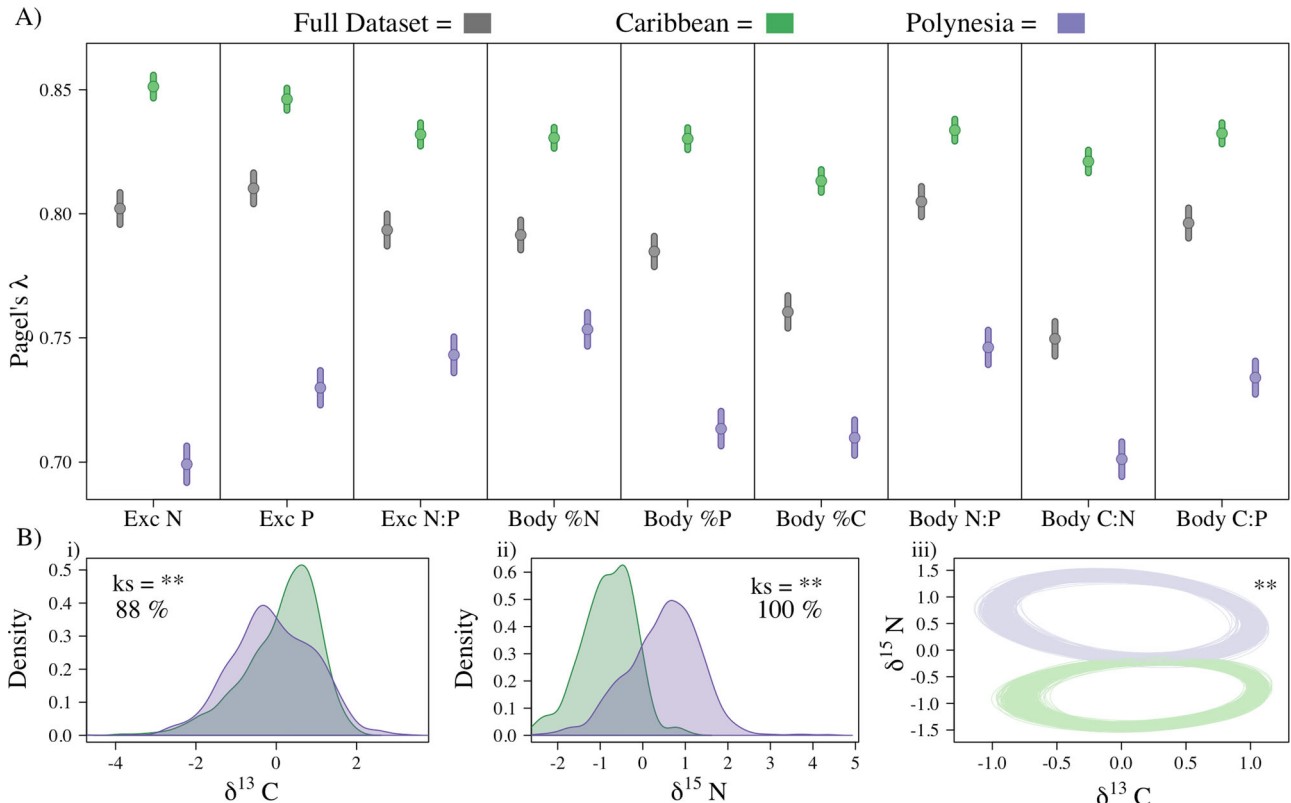

**Fig. 2 The larger phylogenetic signal for all traits is consistent with a more constrained dietary niche in the Caribbean. A** Relative phylogenetic signal for the two regions and the full dataset (gray) as quantified by Pagel's Lambda ($\lambda$) using a bootstrapped analysis for each trait whereby the points represent the mean and error bars indicate SD of all bootstrapped iterations. In all cases, values significantly differ across regions and also between regions and the full dataset. **B**i–ii Comparisons of standardized $\delta^{13}$C and $\delta^{15}$N ($n = 499$ and 829 for Caribbean and Polynesia, respectively for both $\delta^{13}$C and $\delta^{15}$N) distributions between two regions. "ks" indicates Kolmogorov–Smirnov tests for differences in the shape of the two distributions. "**" indicates statistical significance at $\alpha = 0.05$. Percentages indicate the percent of iteratively sampled communities in which statistical significance was confirmed. **B**iii Standard ellipse area estimated from standardized isotopic values generated from iterative resampling (without replacement) of individuals from the two regions. "**" indicates that in all iteratively sampled communities ellipse areas were larger in Polynesia than the Caribbean—note the comparison of niche width is only for relative size, not location of the ellipse. The convex hull was also estimated for each sampled community and for all iteratively sampled communities the convex hull was larger in Polynesia than in the Caribbean.

such as those in The Bahamas should be more P limited[33–35] due to the underlying geology[31].

To understand why species in these regions exhibit different chemical traits beyond that expected by phylogenetic relatedness, we need to understand what aspects of these species' assemblages may be driving this variation across regions. A unique feature of our dataset is that of the 41 families present, 11 are found in both locations, accounting for 62 and 77% of all individuals in the Polynesian and Caribbean datasets, respectively. Previous research has shown that there is little chemical trait variation within families relative to chemical trait variation among families[19,22,23,37,38]. The substantial overlap in con-familial species in our data allowed us to test a second hypothesis: regional differences in chemical traits are driven by differences between con-familial species across the two study locations. Differences in chemical traits exceeding what would be expected based on phylogenetic relatedness would suggest that differences in organismal ecology were playing an important role in shaping chemical trait variation across the regions.

Using models that both did and did not account for phylogenetic signal ("Methods"), we tested: (1) for absolute intra-family differences in chemical traits between regions without accounting for phylogenetic relatedness (ANOVA model), and (2) if differences in chemical traits exceeded what would be expected by a strong phylogenetic signal. Absolute differences in traits within families between regions emerged when not accounting for phylogenetic relatedness (Fig. 3B). However, the phylogenetically corrected analysis showed that these absolute differences were not beyond expectations based on phylogenetic relatedness. In other words, differences in chemical traits did not exceed expected differences based on phylogenetic relatedness alone, obviating the need to invoke shifts in ecology among closely related organisms to explain regional differences (Fig. 3B). Interestingly, despite different outcomes, residuals from each model were very similar, though more constrained around zero when phylogenetically corrected, highlighting that the taxonomic level of family is very good at explaining chemical trait variation in the ANOVA model (Fig. 3C). Taken collectively, these results suggest that: (1) using the full dataset, chemical traits between the two regions differed beyond what was expected based only on phylogenetic relatedness, (2) these differences align with differences in nutrient limitation in our study locations, and (3) these differences are not explained by changes in closely related species – and thus likely did not occur over shorter ecological time periods. These findings provide further support that is consistent across all analyses in our study that chemical trait variation is shaped more by evolutionary history than by ecological factors.

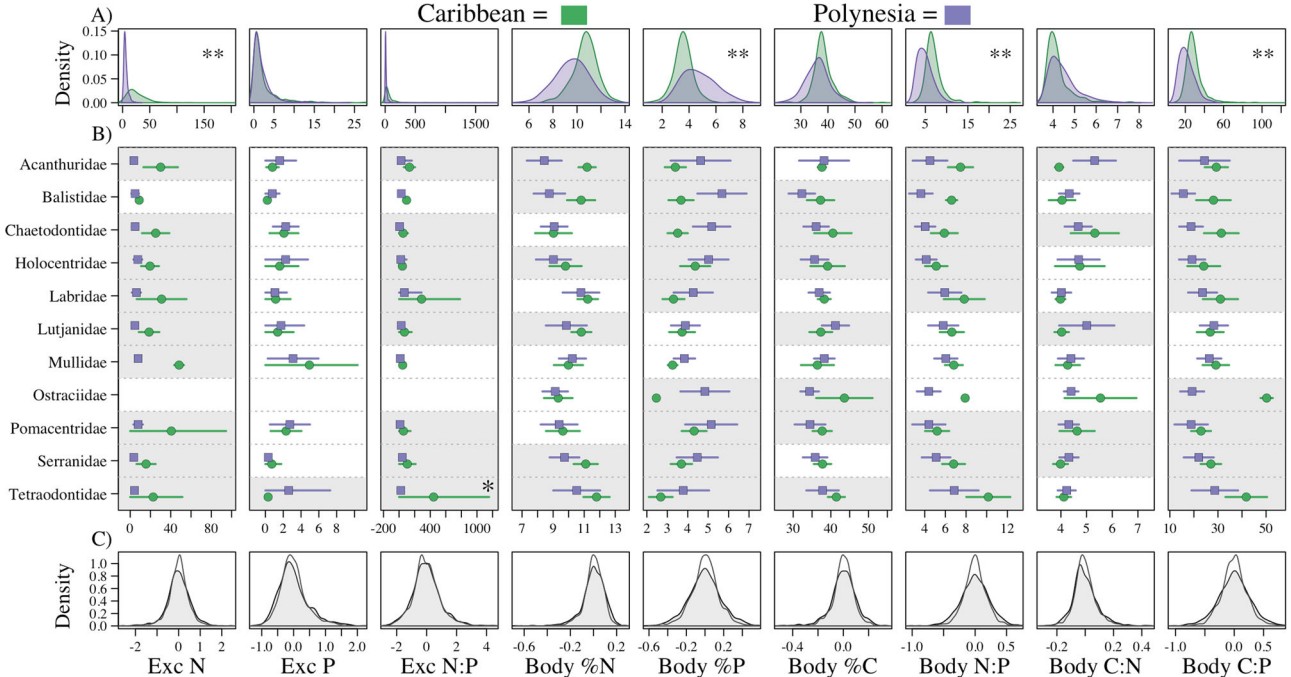

**Fig. 3 Absolute differences in the chemical traits of species that belong to the same family, but live in different regions, are explained by phylogenetic differences, alone. A** Distribution plots of raw data across the Caribbean (green) and Polynesia (purple) for each chemical trait (labels on x-axis of panel **C**). "**"** indicates statistical significance as determined from the global model in Question 1 ($n = 1266, 1133, 1123, 1255, 1255, 1255, 1255, 1255$, and $1255$ individuals, respective to the traits labeled in **C**). **B** Absolute means (symbols) and standard deviations (lines) from raw data of chemical traits for each of the twelve families that are represented in both regions. Chemical trait data points for each family pair with shaded gray backgrounds indicate absolute trait differences quantified by ANOVA. Values associated with "*" indicate trait differences that are beyond what would be predicted by evolutionary history. Empty spaces for Ostraciidae are due to missing data for this family for these traits. **C** Distributions of the residual error from each model (ANOVA = gray, Phylogenetic model = white) for each trait.

## Discussion

Reconciling the factors that predict how species mediate ecological processes is a critical challenge to establishing broadly predictive models of ecological dynamics. Our study shows that variation in nine chemical traits, across a large number and diversity of fishes, is best explained by phylogenetic relatedness as a result of strong phylogenetic conservatism. We did find some evidence for the importance of ecological factors associated with resource acquisition in Polynesia, where phylogenetic conservatism is comparatively lower. We also found that certain chemical traits (Exc N, Body P, Body N:P, Body C:P) differed between regions beyond predictions based on phylogeny alone, providing support that biogeographic differences in nutrient limitation can lead to these differences in chemical traits over evolutionary timescales. However, consistent with previous work that highlights the taxonomic level of family to be the best predictor of chemical trait variation[19,22,23,37,38], we found that differences in chemical traits across regions were not driven by differences within families. Our study demonstrates that evolutionary history is not only an important driver of the traits that mediate species interactions[13,14,39] but is a critical determinant of traits (some of which are themselves ecological processes) that determine dynamics at larger scales of biological organization—the ecosystem.

The ecological factors in our analyses are important indicators of how individuals acquire nutrients or their physiological demand for nutrients, and thus we hypothesized that they would strongly influence chemical traits, particularly excretion[17,37,40]. Yet, we found varied and predominantly weak support for the importance of these ecological factors as predictors of chemical traits. For example, body nutrient demand (e.g., body %N) should predict nutrient excretion, with the expectation that higher

demand results in lower excretion[17]. However, body nutrient demand was not a good predictor of excretion rates. Body nutrient demand was a consistently good predictor of other body nutrients, which provides additional support for previously demonstrated relationships between important macronutrients, e.g., organisms with more body C (and thus higher C demand) tend to have less body P (and thus lower P demand)[22,41]. Beyond body nutrient demand, the diet should be a good predictor of chemical traits, but we found that trophic groups, representing a categorical variable that indicates relative diet quality, were generally poor predictors of all traits. In contrast, continuous measures of diet ($\delta^{13}C$—resource breadth, and $\delta^{15}N$—food chain length) were among the best predictors, but more so for body nutrients than excretion. That the continuous measures of diet were good predictors of body nutrients provides some support for the idea that organisms should minimize dietary mismatches, e.g., herbivores feeding on nutrient-poor diets should have low body nutrient content; a tenet of Ecological Stoichiometry Theory (EST[17]). However, our data were somewhat equivocal with respect to theory: organisms higher on the food chain tended to have lower C:P, lower N:P, and higher %P (in support of EST), but lower %N, and higher C:N (in contrast with EST). Nonetheless, the relatively weak support for ecological factors as predictors of body nutrient demand suggests that nutritional demand at ecological timescales is not a strong driver of chemical trait variation.

Reduced phylogenetic trait conservatism may be expected when competitive interactions among closely related species are more intense since competition can lead to evolutionary divergence via ecological character displacement[42,43] (though the generality of this is contentious[44], particularly with the longer timescales relevant to our study[45]). Our data may provide indirect

evidence of this relationship. Fishes in the Polynesian fish assemblage tend to be slightly closer related than those in the Caribbean (mean pairwise phylogenetic distance was 7% shorter in Polynesia), but are more speciose, and thus Polynesia has more species partitioning the energetic base of coral reef food webs. Interestingly, phylogenetic conservatism was substantially weaker in Polynesia, and this coincided with longer food chains, wider resource breadth, and broader niche widths (Fig. 2B)—all indicating diversified means of resource acquisition consistent with ecological character displacement. This is further supported by the fact that resource breadth and food chain length ($\delta^{13}$C and $\delta^{15}$N, respectively) were among the most consistently significant and strongest ecological factors in the phylogenetic models from Question 1. Taken collectively, these findings suggest that although ecological factors associated with nutritional acquisition at ecological time scales may not strongly influence chemical trait variation, ecological factors that promote competition and species divergence at longer time scales can subsequently give rise to chemical trait variation that is relevant for contemporary ecological dynamics.

If factors that influence nutrient acquisition on ecological scales are poor predictors of chemical traits, limitations in the environmental availability of nutrients, e.g., associated with underlying geology, may have important implications on longer timescales. A key strength of our study was that in addition to the diversification that occurred across the different regions, the locations from which these fishes were collected have fundamentally different geologic histories and thus different availabilities of essential nutrients. The Bahamas are carbonate island formations of coral reefs[46], resulting in an extreme limitation in P availability[33–35] (P physically adsorbs to carbonate). In contrast, the P-rich volcanic islands of Polynesia, in particular high-islands like Mo'orea, are considered more N limited[31,32]. P is essential for bone and scale development in fishes and, in accordance with relative availability, body P was higher in Mo'orea than in The Bahamas while body C:P and N:P were lower (Fig. 3A). Further, in accordance with N being more limiting in Mo'orea, N excretion rate was significantly lower in Mo'orea, indicating that due to the high demand of N, these fishes were assimilating greater amounts of N from their diets (body N was also absolutely lower in Mo'orea but not beyond that predicted by phylogenetic relatedness). Importantly, we also show that the cross-region differences in chemical traits cannot be explained by trait differences within families found in both regions (e.g., Lutjanidae species in the Caribbean versus Polynesia). This, taken with consistent findings that families explain the most variation in chemical traits[19,22,23,37,38], suggests that regional differences may be driven by families unique to each region. We acknowledge that we cannot speak to within-region biogeographic patterns because our data were collected from specific locations (one island in each region) as opposed to more broadly across these regions. However, our study raises important questions regarding the implications of ecosystem nutrient limitation for chemical traits among organisms within the same region, e.g., The Bahamas (a low-island) versus Cuba (a high-island) in the Caribbean. Still, our findings are remarkably consistent with first principle expectations of nutrient physiology and mass balance, and provide novel support for a fundamental premise about consumer nutrient traits: nutrient availability has a strong influence on chemical traits, but this is largely evident across longer, evolutionary timescales.

Ecological and evolutionary processes interact to drive trait variation among species. Yet, because of the separate scales at which ecological and evolutionary processes are perceived to operate, the importance of evolution is often overlooked for its role in contemporary ecological dynamics—particularly among ecosystem ecologists who are largely concerned with the flow of

energy and nutrients. On coral reefs, chemical traits are the means by which fishes store and supply nutrients, and these nutrients are important for fueling the exceptionally high rates of productivity found in these ecosystems[28]. We show through multiple lines of evidence that evolutionary history, and specifically phylogenetic conservatism, of coral reef fishes in two diverse regions, is by far the strongest predictor of chemical traits, and thus represents an important mechanism that drives contemporary nutrient dynamics on coral reefs. The significance of these findings is that they provide strong evidence that the identity of organisms, as a function of their evolutionary history, more so than their ecology, is a strong and useful predictor of ecosystem processes.

## Methods

**Fish capture**. Individual organisms were collected using hook and line, traps, cast nets, and dip nets between 2008 and 2011, within the same large embayment (the Bight of Old Robinson) on Abaco, The Bahamas[23,28], and using barrier nets, dip nets, cast nets, traps, and hook and line in 2016 and 2017 in Moorea, French Polynesia. Captured individuals were immediately transported back to shore in an aerated cooler. Bioassays were conducted on live fish to measure excretion rates of nitrogen (Exc N) and phosphorus (Exc P). Fish were euthanized, stomach contents were removed, and then frozen for transport to the University of Georgia (UGA; Caribbean fishes) or the University of California, Santa Barbara (UCSB; Polynesian fishes) where they were lyophilized, ground, and processed for P by the authors, or were sent to UGA's Isotope lab for quantification of carbon (C), nitrogen (N) content and $\delta^{13}$C and $\delta^{15}$N (see SI). The capture and handling of fish for this project complied with all relevant ethical regulations for animal testing and research at the University of Georgia's Institutional Animal Care and Use Committee (in The Bahamas; AUP # A2009-10003-0), and the University of California Santa Barbara's Institutional Animal Care and Use Committee (in Mo'orea; IACUC #915 2016–2019). In total, we measured nine chemical traits on 1525 individual fishes representing 178 species from 41 families across two coral reef regions (805 individuals in 107 species from ~1300 total potential species[47], and 656 individuals in 71 species from ~280 total potential species[48] in Polynesia and the Caribbean, respectively; Supplementary Table 1).

**Excretion experiments and quantifying body nutrient content**. Excretion rates, for nitrogen ($NH_4+$) and phosphorus (soluble reactive phosphorus; SRP), were measured following the methods of refs. [23,28] as amended in refs. [49,50]. Individual fish were placed in an incubation chamber (0.47–75 L Ziploc bag) containing a known volume (0.08–19.5 L) of prefiltered seawater (0.7 μm pore size Gelman GFF) for 30 min. Experiments were conducted in situ, and bags were placed in bins of ambient seawater. As such, the temperature was highly regulated for all experiments and was similar to the temperature of the environment from which the subject animals were extracted (22–23 °C in The Bahamas, 25–27.5 °C in Moorea). The volume of water per experiment varied according to the size of the individual (0.15–22 L) and net excretion rates were corrected for water volume to achieve a rate of excretion in grams of nutrient per unit time. Values were control corrected through the use of multiple (typically $n = 6$) identical control incubation bags without fish. Each individual used for excretion experiments ($n = 656$ individual fishes, size range: 0.1–2597 g in the Caribbean, and $n = 805$ individual fishes, size range: 0.1–1250 g in Polynesia) was weighed for wet mass and measured to standard length. Fishes were dissected to remove stomach contents and, and after identification, were frozen for transport to UGA (Caribbean fishes) or UCSB (Polynesia fishes) and processed for elemental content. Water samples (filtered with 0.45 μm Whatman nylon membrane filters) were immediately placed on ice and, within 10 h, analyzed for $NH_4+$ using the methodologies of ref. [51] by JEA in The Bahamas, or by KSM and JEA in Mo'orea, or frozen for transport to UGA for SRP analysis using the ascorbic acid method and colorimetric analyses[28,52], by JEA for the samples from The Bahamas, or by KSM and JEA for those from Mo'orea.

Individuals used for somatic nutrient content analyses ($n = 656$ and 805 from the Caribbean, and Polynesia, respectively) were weighed for wet mass and measured to standard length. Samples were lyophilized to a consistent dry weight then ground to a powder with a ball mill grinder. Larger individuals required blending to homogeneity before mill grinding. Ground samples were analyzed for %C and N content and $\delta^{15}$N and $\delta^{13}$C with a CHN Carlo-Erba (NA1500) CN Analyzer, and for %P using dry oxidation-acid hydrolysis extraction followed by colorimetric analysis (Aplkem RF300). Elemental content was calculated on a dry weight basis. All ratios for both body nutrients and excretion are molar.

## Statistical analyses

*Response and predictor variables*. We were interested in the relative chemical trait per individual (response variables), and used mass-corrected measures for all traits—body chemistry is represented as percent nutrient per dry body mass (e.g., %P) and excretion rates are represented as grams of nutrient per time per wet body

mass ($mg_{nutrient}$ $min^{-1}$ $g^{-1}$)(see SI for results showing the same outcomes as presented in the main text using excretion data that is not mass-corrected). The predictor variables included terms for course TG (herbivore, invertivore, omnivore, and predator[28,53]), natural abundance of stable isotopes ($\delta^{13}C$ and $\delta^{15}N$; "Methods"), body mass, and body nutrient demand (Body %N, P, C; "Methods"), and region (Caribbean or Polynesia).

**To what extent is trait variation explained by ecological factors or phylogenetic relatedness?** Bayesian phylogenetic models allowed us to estimate the relative importance of the ecological factors associated with contemporary nutrient acquisition (fixed effects) and simultaneously quantify the relative extent to which each trait was explained by phylogenetic relatedness among species using a species-level intercept with the covariance matrix of phylogenetic distances between species. To estimate phylogenetic relatedness, we used a time tree that only included species placed with genetic data[54] obtained with the "fishtree" package in R[55]. The phylogeny was made ultrametric using non-negative least squares[56] and then was converted into an inverted phylogenetic covariance matrix using the algorithm from[57] implemented with the "inverseA" function from the MCMCglmm package in R[58]. This covariance matrix was then incorporated as a random effect in the Bayesian phylogenetic mixed model. Models were run in R using the "MCMCglmm" package[58]. We used uninformative priors and ran the model for 80,000 iterations with a burn-in of 10000. Model convergence was assessed by inspecting trace plots. We quantified the total variation in the data explained by (1) a "full model" that included all fixed effects (ecological factors) plus a random intercept (species-level effects) from which we partitioned the amount of variance explained by the ecological factors alone and the species-level random effects (phylogenetic effects), and (2) the "phylogeny-only", or intercept-only model, that only included the phylogenetic covariance error structure following[59]. To compare models, we quantified DIC for each model[60].

**How does the relative strength of phylogenetic conservatism differ between the Caribbean and Polynesia?** The strength of phylogenetic conservatism, or the phylogenetic signal, was determined by calculating Pagel's Lambda and Bloomberg's K using the *phylosig* function in the "picante" package in R, including the randomization test for the statistical difference using 999 simulations[61]. Calculating these values requires species-level means. Given the wide range of individual data points we have per species (1–115 per species), we used a bootstrapping procedure whereby we iteratively resampled (creating each time resampled data of equal size to the observed data) with replacement from each region and the full dataset to calculate the species-level mean 1000 times, each time recalculating Pagel's Lambda and Bloomberg's K based on this resampled estimate.

*Isotopic niche space.* We were interested in comparing the relative community-level resource breadth, food chain length, and dietary niche breadth. To do this we quantified four measures: the relative distribution of $\delta^{13}C$, $\delta^{15}N$, and the SEA and Convex hull[62], respectively. Since isotopic baselines are likely different in these two regions, we standardized (*z*-score) all isotopic data within each region prior to analysis to allow for relative comparisons to be made. This is justifiable for four reasons: (1) the relative length of the food chain is comparable because the species among the two regions are all fishes and are relatively closely related, thus similar isotopic fractionation is expected, (2) while basal resources and thus the $\delta^{13}C$ signature may be different in the absolute sense, we are only concerned here with the relative spread of the distribution of these data, (3) for these reasons, the relative size, but not the centroid location, of the isotopic niche width was compared, (4) we used iterative procedures to ensure we were comparing communities of similar size and to account for sampling bias that may have occurred when collecting specimens (this is highly conservative because specimens were collected in the same manner and largely by the same observers in both locations—see plots of species-level isotopes from both regions Supplementary Fig. 3). To generate estimates of resource breadth, food chain length, and dietary niche breadth we iteratively sampled the species in each region without replacement to generate 500 communities of 250 individuals (representing ~50% of the number of individuals in the Caribbean). For each pair of communities, we quantified the dietary niche breadth by calculating the SEA, and the Convex Hull Area using the *standard.ellipse* and *convexhull* functions in the R Package Siar[62]. We quantified resource breadth and food chain length, the spread of $\delta^{13}C$ and $\delta^{15}N$, respectively, and compared the distributions of each isotope using a Kolmogorov–Smirnov test for each iteration. To verify our findings, we conducted the same analyses using ~75%, and 90% of the number of individuals in the Caribbean (375 and 450 individuals in each community, respectively). The only difference in outcomes was that distributions of $\delta^{13}C$ between regions differed 100% of the time (as opposed to ~90% of the time) when the sample size was > %50.

**To what extent does biogeographic history influence trait variation beyond that predicted by phylogenetic divergence?** We first identified which traits differed across regions using the output from Question 1, i.e., the significance of the fixed term "region" in the model (Fig. 3A). We then used two different models to test if these differences were explained by trait variation within families using two different models that examined: (1) the absolute differences in chemical trait values

within families across the region, and (2) differences in these values when accounting for phylogenetic relatedness of species within families. Respectively

(1) Trait ~ 1 + family:region
(2) Trait ~ 1 + family:region + 1 | sp + 1 | phylo

Models 1 and 2 were run using "MCMCglmm" package in R[58], using uninformative priors and for 80,000 iterations with a burn-in of 10,000. Model convergence was assessed by inspecting trace plots.

**Reporting summary.** Further information on research design is available in the Nature Research Reporting Summary linked to this article.

## Data availability
Public availability of data is currently pending but can be made available to individuals upon request from Jacob Allgeier, contact: jeallg@umich.edu.

## Code availability
Code for all analyses and figures is available at https://github.com/Allgeier-Lab/Allgeier_et_al_2021_NatureCommunications.

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

## Acknowledgements

We would like to thank the country and people of The Bahamas for allowing continuous research in their beautiful waters and to local fishers who provided countless hours of local knowledge of where and how to fish, local traps, and conch slop swaps for Guinness that were essential for collecting many of the fishes within this dataset. We would like to thank the country of French Polynesia and the people of Mo'orea for allowing the continual study of their marine ecosystems and to local community members, notably Moana at the corner shop, for helping us identify locations and methodologies for the collection of many species within this dataset. We would like to thank Craig Layman, Amy Rosemond, Friends of the Environment, Darlene Haines, Jim Richard, Richard Appaldo, and Nathalie Giroire for their tremendous support without which the research in The Bahamas could not have been accomplished. We would like to thank Tom Maddox of the Analytical Chemistry Lab, the University of George who was instrumental in chemical analysis mentorship and processing. We would like to thank Mark Hunter, Sean Geary, and members of the Allgeier lab, and three reviewers for thoughtful comments that greatly improved the manuscript. We would like to thank the Mo'orea Coral Reef Ecosystem LTER, funded by the US National Science Foundation (OCE-1637396 and earlier awards). Support for this study was provided by Lucille and David Packard Fellowship and National Science Foundation OCE #1948622 to JEA, Mo'orea Coral Reef LTER (OCE-1637396, and previous grants), and National Science Foundation OCE-1547952 to D.E.B.

## Author contributions

J.E.A. designed the study and conducted analyses with important contributions from B.C.W., N.W., S.J.W. and D.E.B., as well as important additional contributions from D.E.B., K.S.M., V.P., N.M.D.S. and S.V. Fieldwork, bioassays, and laboratory work was conducted by J.E.A., K.S.M. and D.E.B. J.E.A. wrote the paper with important contributions from B.C.W., K.S.M., N.W., S.J.W., D.E.B., V.P., N.M.D.S. and S.V.

## Competing interests

The authors declare no competing interests.
