## [Peer Review File · Nature Communications]

Phylogenetic conservatism drives nutrient dynamics of coral reef fishesREVIEWER COMMENTS

Reviewer #1 (Remarks to the Author):

General comments

The manuscript from Allgeier et al. investigates the relative importance of ecological factors and evolutionary relatedness underpinning differences in traits associated with nutrient dynamics (what they call 'chemical traits') in coral reef fishes. By applying a comparative approach across two biogeographically distinct locations (Caribbean and Polynesia) they find that phylogenetic history outweighs other ecological factors in predicting species chemical traits. Additionally, they perform explicit comparisons between biogeographic locations to show that the Caribbean has a stronger phylogenetic signal. Finally, they suggest that different nutrient regimes might drive disparities in chemical traits among regions.

Overall, I found this an interesting paper and pleasant to read. It provides important insights about the imprint of evolutionary and biogeographic histories in ecosystem processes, which is something relatively novel. I believe this study will be of general interest, however, I did find some key points that should be addressed before it can be considered for publication. As you will see from my comments below, my main concerns are related to a better description of the Methods to improve clarity and allow reproducibility. However, I also make some suggestions for improvement in other sections of the manuscript. I hope this review is helpful.

Major points

The introduction is quite well written, I like how you set up the broader theoretical framework supporting the study in the first three paragraphs. However, I missed a better contextualization of nutrient dynamics on coral reefs specifying how fish are important in ecosystem processes within those environments. Perhaps just add another short paragraph (or expand the last one) to talk about the specific context of chemical traits in coral reef fishes and their relevance for nutrient dynamics. This will make it easier for the reader to navigate your introduction leading to the objectives of the study.

In your results, you mention that '...the variation explained by the full model (including ecological terms and phylogeny) did not differ from that explained by the phylogeny-only (intercept-only) model' (Lines 121-122). I was curious to see that in some cases, the proportion of variance explained by the full model was slightly lower (with overlapping CIs) than the phylogeny-only model (e.g. Exc N, Exc N:P, body %P). What would explain this if ecology does contribute to the explanation in some of these cases?

On a related note, I noticed that the proportion of variance explained by the phylogeny-only model in 'Exc N:P' was quite low (~20%), and yet the phylogenetic signal for this chemical trait was found to be as high as for the other traits. If you are saying that the high amount of variance explained by relatedness is an indication of phylogenetic conservatism (Lines 126-127), one would assume that 'Exc N:P' should also have a lot of its variance explained by phylogeny, but that was not the case. I am bringing this up

because, as far as I am aware, the high proportion of variance explained by phylogenetic relatedness in hierarchical models does not necessarily equate to phylogenetic signal in the traits, but it means that the residuals present some sort of phylogenetic structure (see Revell 2010 - *Methods in Ecology and Evolution*). I think it would be important to reassess your results with this in mind and, if necessary, reword some parts of the manuscript to avoid confusion.

In the discussion, you raise the point that '...biogeographic differences in nutrient limitation can lead to these differences in chemical traits over evolutionary timescales'. I think this is a very strong statement that assumes that the regions analyzed have been limited by specific nutrients through geological time. It also assumes that species evolved in locations with that specific nutrient limitation. It is important to have in mind that some of the species analyzed have quite large geographic distributions and experience varied environmental characteristics throughout their range. Therefore, I would interpret the results of phylogenetic signal as evidence for species inheriting chemical traits via common ancestry but that nutrient limitation can make differences manifest within the inherited chemical trait range.

Still in the discussion, it was not very clear for me where in your results you found that '...differences in chemical traits across regions were not driven by differences within families, but more by families and species unique to each region'. I might be missing something, but where did you analyze the families unique to each region? It might be necessary to make it more explicit in the results.

I consider the result of higher phylogenetic signal in the Caribbean as a very interesting one, however, I believe it has not been properly discussed. I see some reasons why this would happen and they are directly related to the evolutionary history of the Atlantic. The fact that the Atlantic has a substantially lower species richness when compared to the Indo-Pacific is not only related to its distance to the center of biodiversity, but it also relates to the episodic faunal extinction events driven by environmental change in the Atlantic (particularly after the closure of the Isthmus of Panama; e.g. O'Dea et al., 2007 - *PNAS*). Therefore, I would expect that out of the pool of species within the Atlantic, you would find that lineages are generally more closely related than the species-rich Indo-Pacific. It would thus be very interesting to know if there are differences in the maximum phylogenetic distance between your two assemblages. If that is the case, this should be added to the manuscript. I think that your discussion will benefit from having this historical perspective about the formation of the two assemblages analysed.

Finally, I have some points that need to be clarified in the methods. First, you cite Allgeier et al. (2015) as the reference for measuring excretion rates, however, within that paper you make reference to other original papers for the methodology. Please, cite the original references so the reader can follow the methods without needing to jump between multiple papers.

I also missed a better description of your predictor variables. In that section, you jump from talking about the mass-corrected traits to talking about the models (Lines 316-322) and it gets really confusing. This model part could potentially be moved to the 'Statistical Analyses' section and you could use that space to elaborate more on the predictor variables.

Within the 'Statistical Analyses' section, I missed a better explanation on how the phylogenetic covariance matrix was calculated. Did you use the stochastically-resolved phylogeny or the phylogeny with genetic data only? This is very important because if you are looking at phylogenetic signal, the stochastic imputation of species can have a big impact in the results and yet this information is nowhere to be found in your methods.

Related to the previous point, as far as I am aware, the 'fishtree' R package does not contain functions to calculate the phylogenetic covariance matrix. What model and package did you use to calculate the matrix and why? This information is crucial, so please be more specific.

In line 410, you say 'We first identified which traits differed across regions...' but again you did not specify how this was done.

Lastly, I am sure that the authors are aware that Scaridae is not a family and that parrotfishes are part of the Labridae. Therefore, if family is your taxonomic unit of analysis, you should include parrotfishes within Labridae and reanalyze the data accordingly. I doubt that this change would have any significant impact on the results, but it is important to be consistent with the currently accepted taxonomic (and phylogenetic) status of the group under investigation.

Minor Points

Line 26: 'drivers' instead of 'a driver'.

Line 27: 'We find that'. The 'that' was missing from the sentence.

Line 33: Maybe add 'the geological context of' after 'associated with' to specify.

Line 82: 'at least' is repeated.

Line 130: remove the 'is' from 'as is'.

Line 137: remove the extra 'his'.

Line 152: I suggest changing to '...French Polynesia offer the opportunity...'.

Line 155: 'these four' might be better than 'certain' to specify the chemical traits.

Line 178: I think 'However' would suit better in this case than 'Importantly'.

Line 185: 'these results suggest that' instead of 'we find that'.

Line 199: 'find some evidence for' instead of 'evidence of'.

Line 235: 'among closely related species'. This is an important distinction if you are talking about reduced phylogenetic signal.

Line 254: 'geologic histories'.

Line 405: 'regions differed'.

Figure 1: I suggest changing the colors of 'Exc N' and 'Exc P' not to overlap with the colors in the subsequent figures that represent something different. Also, I think it would facilitate the visualization of the figure if parts A and B were slightly more separated.

Figure 2: You need to explain why there are empty boxes in B for Ostraciidae, maybe in the legend, methods or both. It is important to clarify.

Figure S2: Is there a plot missing from this figure? In the legend you say 'subsequent plot' and describe it, but it is nowhere to be found. Please double check.

References

Revell, L. (2010). Phylogenetic signal and linear regression on species data. *Methods in Ecology and Evolution*, 1, 319–329.

O'Dea, A., Jackson, J. B. C., Fortunato, H., Smith, J. T., D'Croze, L., Johnson, K. G., & Todd, J. A. (2007). Environmental change preceded Caribbean extinction by 2 million years. *PNAS*, 104, 5501–5506.

Reviewer #2 (Remarks to the Author):

Review of Phylogenetic conservatism drives nutrient dynamics of coral reef fishes

I have been asked to review this manuscript as I have general expertise in coral reef evolutionary ecology, but would be considered a non-specialist regarding this particular topic. I consider this type of perspective important for a top-tier general journal, in this case *Nature Communications*, as it is important that papers in these journals (i.e. major contributions) attract citations from similar, but adjacent, fields. However, given my expertise and role as a reviewer in this instance, I will broadly refrain from commenting on the details of the methods and how this manuscript fits in alongside other field-specific literature.

Overall, I think this manuscript needs some work to increase its appeal to non-specialists (e.g. more general ecologists), but it is my opinion that it is of the standard required for serious consideration at *Nature Communications*. The question being addressed, sample size and approach are fantastic. My main concern regards the general pitch, as it currently reads as though it is targeting a much more specialised journal. I think a lot of this can be sorted by giving more detailed examples and more explicitly stating why this study is important and why researchers in similar fields should care – currently it seems as though the discussion on this is not targeted in a way that would make another ecologist (who does not work on this topic) want to cite it in a general way. Maybe asking for comments from colleagues who work on other topics will help.

I suggest revision of the manuscript and have provided some more specific comments below. This is an impressive study and certainly interesting, but just feels to me as though that is not being sufficiently communicated!

L 34: Do these results really suggest that “phylogenetic conservatism is a critical determinant”, or does it suggest that phylogenetic position is a better predictor than ecological factors regarding nutrient dynamics on coral reefs? To me, it seems the latter.

L 43: Maybe this is a bit cosmetic, but I think for people outside this specific field, an actual example would also be helpful here (rather than a description of a function labelled as an example).

L 49: Are nitrogen, phosphorous and carbon “chemical traits”? That seems like an odd description to me.

L 56-57: This example (and especially that from 59-62) is better than that on 43, as it allows a non-specialist in this particular field to appreciate exactly what’s going on.

L 108: Why wasn’t “geographic location” (Moorea vs Caribbean) included as a fixed effect? It would also be informative to have an idea of the phylogenetic and “ecological” diversity of the species sampled from the Caribbean and Moorea relative to the species that exist in those regions – i.e. how skewed were the data? I understand this was addressed in latter parts of the study, but why was it excluded here as it might help explain the results?

L 165: Might be a dim question, but how is there a range for 62-77% - shouldn’t that be one number?

L 185 and L 190 both start with “collectively”

L 197: I would suggest you remove the word “exceptional” – too emotive

Reviewer #3 (Remarks to the Author):

Allegeir et al. set out to explore the factors that determine organismal chemical traits across two distinct geographic regions. Specifically, they explore if phylogenetic relatedness or various ecological factors drive fish chemical traits. The authors do a nice job bringing in both evolutionary and ecological literature and work to bridge these fields in this study. In comparison to some of these authors' previous work (Allegeir et al. 2020), I find these analyses to be more robust and convincing. I thought this was a strong study and was strengthened by the use of communities from two distinct regions with unique geologies and background nutrient limitation status.

The article provides novel results and the conclusions are backed up by the data. While I thought the analyses were appropriate, I think this paper would better highlight the phylogenetic conservatism suggested here by providing a phylogenetic tree of the study species from the two regions and somehow highlight or map the traits onto this tree to better visualize how these traits are conserved across the evolutionary lineages. This may not need to be in the main text, but included as a supplement. However, I think this would better highlight the relatedness amongst these families and how the traits are actually conserved across time. With that, is there a dated phylogeny of these species as suggested in the text regarding the time since divergence?

Some more specific comments are highlighted below.

In regards to the methods, did the authors note any inter-annual variation in the traits they measured? For example, different chemical analyses were used in different years.

Line 360: Can more details be provided regarding the phylogenetic covariance matrix? For example, are these maximum likelihood or Bayesian covariance matrices?

Line 384: Can the authors be more clear how isotope values were standardized? Are they simply stating since they did not examine the centroid and only the distribution of data (i.e., amount of area or variance) that is how it was standardized? This could be more clear.

In figure S2, I found it curious that the communities with the higher K-statistic and Pagel's values were less-likely (as indicated by the proportions above) to be different from a randomly simulated community. What is driving that? Also, the values at the top appear to be percentages as opposed to proportions.

What do the authors mean by phylogenetic history on line 27? Phylogenetic relatedness would be more accurate.

Lines 90-91: Are there citations for these statements regarding nutrient limitation differences in the two study regions?

Line 201: What chemical traits? Be specific. It would be helpful to define which ones varied.

Reviewer #1 (Remarks to the Author):

General comments

The manuscript from Allgeier et al. investigates the relative importance of ecological factors and evolutionary relatedness underpinning differences in traits associated with nutrient dynamics (what they call ‘chemical traits’) in coral reef fishes. By applying a comparative approach across two biogeographically distinct locations (Caribbean and Polynesia) they find that phylogenetic history outweighs other ecological factors in predicting species chemical traits. Additionally, they perform explicit comparisons between biogeographic locations to show that the Caribbean has a stronger phylogenetic signal. Finally, they suggest that different nutrient regimes might drive disparities in chemical traits among regions.

Overall, I found this an interesting paper and pleasant to read. It provides important insights about the imprint of evolutionary and biogeographic histories in ecosystem processes, which is something relatively novel. I believe this study will be of general interest, however, I did find some key points that should be addressed before it can be considered for publication. As you will see from my comments below, my main concerns are related to a better description of the Methods to improve clarity and allow reproducibility. However, I also make some suggestions for improvement in other sections of the manuscript. I hope this review is helpful.

Major points

The introduction is quite well written, I like how you set up the broader theoretical framework supporting the study in the first three paragraphs. However, I missed a better contextualization of nutrient dynamics on coral reefs specifying how fish are important in ecosystem processes within those environments. Perhaps just add another short paragraph (or expand the last one) to talk about the specific context of chemical traits in coral reef fishes and their relevance for nutrient dynamics. This will make it easier for the reader to navigate your introduction leading to the objectives of the study.

Thank you for your positive comments about the introduction. We have added text speaking to the importance of fishes for mediating nutrient dynamics on coral reefs (Lines 92-97).

In your results, you mention that ‘...the variation explained by the full model (including ecological terms and phylogeny) did not differ from that explained by the phylogeny-only (intercept-only) model’ (Lines 121-122). I was curious to see that in some cases, the proportion of variance explained by the full model was slightly lower (with overlapping CIs) than the phylogeny-only model (e.g. Exc N, Exc N:P, body %P). What would explain this if ecology does contribute to the explanation in some of these cases?

We agree this is a bit counterintuitive. Calculating R^2 in mixed-effects models, in our case a GLMM, is known to be challenging, primarily because by definition mixed-effects models have more than one variance component (i.e., from the error term and the random effect), in contrast to traditional linear models that have only one (Nakagawa and Schielzeth 2013, Nakagawa et al. 2017, Ives 2019). Nakagawa and Schielzeth 2013 note that a primary concern is that in mixed models the R^2 value can decrease as the number of terms in the model increases in sharp

contrast to a traditional R^2 from and linear model that can only increase with increased number of terms. An updated formulation of R^2 for mixed-models presented by Nakagawa and Schielzeth 2013 involves generating two types of R^2 , the marginal and conditional, that explain variance associated with the fixed effects only, and the fixed+random effects, respectively. Doing so reduced the probability of generating a reduced R^2 with increased number of terms, but does not fully eliminated this issue (Nakagawa and Schielzeth 2013). In our study, we calculated the marginal R^2 to quantify the explanatory power of the fixed effects (what we call our ‘ecology’ terms) and the conditional R^2 to quantify the explanatory power of the full model (fixed + random, or ecology + phylogeny) using an adjusted formulation provided by Nakagawa and Schielzeth 2013 for the Bayesian MCMCGLmm model output (see <https://stat.ethz.ch/pipermail/r-sig-mixed-models/2015q3/023861.html>). In this case comparison between the marginal R^2 and conditional R^2 is appropriate and in all cases with our analysis, showed that the ecology terms do not explain a substantial proportion of the variation in the data. We then ran a separate intercept-only model for which we calculated conditional R^2 only, which in this case given there are no fixed effects represents variation in the data explained only by the random effect of phylogeny. The reviewer correctly noted that in many cases the intercept-only model explained more variation in the data than the full model; the issues of accounting for multiple variance components is one explanation for this. We have amended the main text to emphasis comparisons between the marginal and conditional R^2 in the full model, and only refer to the intercept-only model to draw attention to the fact that it alone explains a high proportion of the variation in the data. Lines 135-153

*Ives, A. R. 2019. R^2 for Correlated Data: Phylogenetic Models, LMMs, and GLMMs. *Systematic Biology* 68:234–251.*

*Nakagawa, S., P. C. D. Johnson, and H. Schielzeth. 2017. The coefficient of determination R^2 and intra-class correlation coefficient from generalized linear mixed-effects models revisited and expanded. *Journal of The Royal Society Interface* 14:20170213.*

*Nakagawa, S., and H. Schielzeth. 2013. A general and simple method for obtaining R^2 from generalized linear mixed-effects models. *Methods in Ecology and Evolution* 4:133–142.*

On a related note, I noticed that the proportion of variance explained by the phylogeny-only model in ‘Exc N:P’ was quite low (~20%), and yet the phylogenetic signal for this chemical trait was found to be as high as for the other traits. If you are saying that the high amount of variance explained by relatedness is an indication of phylogenetic conservatism (Lines 126-127), one would assume that ‘Exc N:P’ should also have a lot of its variance explained by phylogeny, but that was not the case. I am bringing this up because, as far as I am aware, the high proportion of variance explained by phylogenetic relatedness in hierarchical models does not necessarily equate to phylogenetic signal in the traits, but it means that the residuals present some sort of phylogenetic structure (see Revell 2010 - *Methods in Ecology and Evolution*). I think it would be important to reassess your results with this in mind and, if necessary, reword some parts of the manuscript to avoid confusion.

Thank you for pointing this out. The reviewers are correct, high proportion of variance explained by phylogenetic relatedness does not necessarily relate directly to phylogenetic signal. Phylogenetic signal can be calculated from the output of the hierarchical model, but we opted to

use a more traditional approach using Pagel's Lambda and Bloomberg K-statistic. We interpret the collective findings of strong support for phylogeny in the hierarchical models and relatively strong phylogenetic signal from the Pagel's Lambda and Bloomberg K-statistic as a demonstration that the traits we were analyzing are phylogenetically conserved. Given these comments by the reviewer, we now see how we have blurred the distinctions between the two analyses and we have omitted specific text about phylogenetic signal in reference to the hierarchical models (Line 136) and (re)added text defining phylogenetic signal (Lines 157-158) to make this distinction more clear.

In the discussion, you raise the point that '...biogeographic differences in nutrient limitation can lead to these differences in chemical traits over evolutionary timescales'. I think this is a very strong statement that assumes that the regions analyzed have been limited by specific nutrients through geological time. It also assumes that species evolved in locations with that specific nutrient limitation. It is important to have in mind that some of the species analyzed have quite large geographic distributions and experience varied environmental characteristics throughout their range. Therefore, I would interpret the results of phylogenetic signal as evidence for species inheriting chemical traits via common ancestry but that nutrient limitation can make differences manifest within the inherited chemical trait range.

Still in the discussion, it was not very clear for me where in your results you found that '...differences in chemical traits across regions were not driven by differences within families, but more by families and species unique to each region'. I might be missing something, but where did you analyze the families unique to each region? It might be necessary to make it more explicit in the results.

We agree with the reviewer that this text is somewhat difficult to interpret. To be clear, we did not conduct any analysis that only included families unique to each region. We conducted analyses that (1) included all families in both regions (Question 1), and (2) included families that were shared across regions (Question 3). This latter analysis showed that differences among regions were not due to within-family differences. From this result one could infer that the remaining differences are thus from the remaining families that are different among the regions. However, to the reviewer's point, we did not make this test per se and thus to clarify we have simply omitted the text "but more by families and species unique to each region" from this sentence. Line 265

I consider the result of higher phylogenetic signal in the Caribbean as a very interesting one, however, I believe it has not been properly discussed. I see some reasons why this would happen and they are directly related to the evolutionary history of the Atlantic. The fact that the Atlantic has a substantially lower species richness when compared to the Indo-Pacific is not only related to its distance to the center of biodiversity, but it also relates to the episodic faunal extinction events driven by environmental change in the Atlantic (particularly after the closure of the Isthmus of Panama; e.g. O'Dea et al., 2007 - PNAS). Therefore, I would expect that out of the pool of species within the Atlantic, you would find that lineages are generally more closely related than the species-rich Indo-Pacific. It would thus be very interesting to know if there are differences in the maximum phylogenetic distance between your two assemblages. If that is the case, this should be added to the manuscript. I think that your discussion will benefit from having this historical perspective about the formation of the two assemblages analysed.

You raise a very interesting question: If, as you reasonably hypothesize, the crown age of the Atlantic assemblage is younger and the species in the Atlantic are, on average, more closely related, could that drive the increased phylogenetic signal we see in the Caribbean?

When we look at the data, we find that: (1) the crown age of the Caribbean assemblage is actually older than the crown age of the Pacific assemblage (patristic distances of the branch lengths from the root to the tips of the ultrametric trees are 146 for the Pacific and 192 for the Atlantic), and (2) the mean-pairwise phylogenetic distances are quite similar, though are slightly larger in the Atlantic (patristic branch length distance of 236) than the Pacific (patristic branch length distance of 220).

These findings suggest that the episodic extinctions in the Atlantic likely cannot be directly linked to the higher phylogenetic signal in our data via the pathway that you outline. However, we agree a caveat that recognizes the potential for different macroevolutionary histories to impact phylogenetic trait conservatism is warranted and will improve the discussion, and we have added text to this effect into the discussion. Lines 298-300

Finally, I have some points that need to be clarified in the methods. First, you cite Allgeier et al. (2015) as the reference for measuring excretion rates, however, within that paper you make reference to other original papers for the methodology. Please, cite the original references so the reader can follow the methods without needing to jump between multiple papers.

We used the citation from Allgeier et al. 2015 only because in this manuscript, we altered the methods from Schaus et al. (1997) and Whiles et al. (2009). Nonetheless, we appreciate the utility of having the most basal papers cited and have included these citations as well making clear these were the original studies and we amended out work from them. Line 379

*Allgeier, J. E., S. J. Wenger, D. E. Schindler, A. D. Rosemond, and C. A. Layman. 2015. Metabolic theory and taxonomic identity predict nutrient cycling in a diverse food web. *Proceedings of the National Academy of Science* 112:2640–2647.*

*Schaus, M. H., M. J. Vanni, T. E. Wissing, M. T. Bremigan, J. E. Garvey, and R. A. Stein. 1997. Nitrogen and phosphorous excretion by detritivorous gizzard shad in a reservoir ecosystem. *Limnology and Oceanography* 42:1386–1397.*

*Whiles, M. R., A. D. Hury, B. W. Taylor, and J. D. Reeve. 2009. Influence of handling stress and fasting on estimates of ammonium excretion by tadpoles and fish: recommendations for designing excretion experiments. *Limnology and Oceanography-Methods* 7:1–7.*

I also missed a better description of your predictor variables. In that section, you jump from talking about the mass-corrected traits to talking about the models (Lines 316-322) and it gets really confusing. This model part could potentially be moved to the ‘Statistical Analyses’ section and you could use that space to elaborate more on the predictor variables.

We agree that the placement of this section was a bit odd and have moved it to “Statistical Analyses”. We have also added further detail regarding the predictor variables in this section. Lines 430-438

Within the ‘Statistical Analyses’ section, I missed a better explanation on how the phylogenetic covariance matrix was calculated. Did you use the stochastically-resolved phylogeny or the phylogeny with genetic data only? This is very important because if you are looking at phylogenetic signal, the stochastic imputation of species can have a big impact in the results and yet this information is nowhere to be found in your methods.

We agree that this is an important point. We used the phylogeny that only included species with genetic data, not the phylogeny that included taxa placed using stochastic polytomy resolution. We have clarified this in the manuscript by expanding the description of the phylogeny and referencing the paper that published the phylogeny using species with genetic data only. Lines 446-452

Related to the previous point, as far as I am aware, the ‘fishtree’ R package does not contain functions to calculate the phylogenetic covariance matrix. What model and package did you use to calculate the matrix and why? This information is crucial, so please be more specific.

Thank you for highlighting this ambiguity; we agree that this is important information to include, and you are correct, we retrieved the phylogeny using ‘fishtree’, but converted it to a phylogenetic covariance matrix using the ‘inverseA’ function in MCMCglmm. This function implements the Hadfield and Nakagawa (2010) algorithm for calculating an inverted phylogenetic covariance matrix from a phylogeny. We have clarified this in the text, and now reference the Hadfield and Nakagawa (2010) paper that discusses the development and justification for use of their algorithm to calculate phylogenetic covariance matrices when fitting phylogenetic mixed models. Lines 446-452

*Hadfield, J. D., and S. Nakagawa. 2010. General quantitative genetic methods for comparative biology: phylogenies, taxonomies and multi-trait models for continuous and categorical characters. *Journal of Evolutionary Biology* 23:494–508.*

In line 410, you say ‘We first identified which traits differed across regions...’ but again you did not specify how this was done.

We have added text to clarify that this was a result that was generated from the model output from Question 1. Lines 507-508

Lastly, I am sure that the authors are aware that Scaridae is not a family and that parrotfishes are part of the Labridae. Therefore, if family is your taxonomic unit of analysis, you should include parrotfishes within Labridae and reanalyze the data accordingly. I doubt that this change would have any significant impact on the results, but it is important to be consistent with the currently accepted taxonomic (and phylogenetic) status of the group under investigation.

Thank you for catching this, we indeed overlooked this and have changed the figure. There were no significant changes needed to be made in the results beyond changing the number of families that were in both regions from 12 to 11.

Minor Points

Line 26: 'drivers' instead of 'a driver'.

This was changed.

Line 27: 'We find that'. The 'that' was missing from the sentence.

This was changed.

Line 33: Maybe add 'the geological context of' after 'associated with' to specify.

This was added.

Line 82: 'at least' is repeated.

This was removed.

Line 130: remove the 'is' from 'as is'.

This was removed.

Line 137: remove the extra 'his'.

This was removed.

Line 152: I suggest changing to '...French Polynesia offer the opportunity...'

This was removed.

Line 155: 'these four' might be better than 'certain' to specify the chemical traits.

This was changed as suggested.

Line 178: I think 'However' would suit better in this case than 'Importantly'.

This was changed.

Line 185: 'these results suggest that' instead of 'we find that'.

This was changed.

Line 199: 'find some evidence for' instead of 'evidence of'.

This was changed as suggested.

Line 235: 'among closely related species'. This is an important distinction if you are talking about reduced phylogenetic signal.

We agree, thank you for catching this. This was changed as suggested.

Line 254: 'geologic histories'.

This was changed as suggested.

Line 405: 'regions differed'.

This was changed as suggested.

Figure 1: I suggest changing the colors of ‘Exc N’ and ‘Exc P’ not to overlap with the colors in the subsequent figures that represent something different. Also, I think it would facilitate the visualization of the figure if parts A and B were slightly more separated.

The colors in the figure were changed as suggested, but the figure was not spaced as suggested.

Figure 3: You need to explain why there are empty boxes in B for Ostraciidae, maybe in the legend, methods or both. It is important to clarify.

Thank you for catching this. We have added text for clarity.

Figure S2: Is there a plot missing from this figure? In the legend you say ‘subsequent plot’ and describe it, but it is nowhere to be found. Please double check.

Thank you for catching this. Apologies, this was a hold-over from an earlier iteration of the MS. This text has been omitted.

References

Revell, L. (2010). Phylogenetic signal and linear regression on species data. *Methods in Ecology and Evolution*, 1, 319–329.

O’Dea, A., Jackson, J. B. C., Fortunato, H., Smith, J. T., D’Croze, L., Johnson, K. G., & Todd, J. A. (2007). Environmental change preceded Caribbean extinction by 2 million years. *PNAS*, 104, 5501–5506.

Reviewer #2 (Remarks to the Author):

Review of Phylogenetic conservatism drives nutrient dynamics of coral reef fishes

I have been asked to review this manuscript as I have general expertise in coral reef evolutionary ecology, but would be considered a non-specialist regarding this particular topic. I consider this type of perspective important for a top-tier general journal, in this case *Nature Communications*, as it is important that papers in these journals (i.e. major contributions) attract citations from similar, but adjacent, fields. However, given my expertise and role as a reviewer in this instance, I will broadly refrain from commenting on the details of the methods and how this manuscript fits in alongside other field-specific literature.

Overall, I think this manuscript needs some work to increase its appeal to non-specialists (e.g. more general ecologists), but it is my opinion that it is of the standard required for serious consideration at *Nature Communications*. The question being addressed, sample size and approach are fantastic. My main concern regards the general pitch, as it currently reads as though it is targeting a much more specialised journal. I think a lot of this can be sorted by giving more detailed examples and more explicitly stating why this study is important and why researchers in similar fields should care – currently it seems as though the discussion on this is not targeted in a way that would make another ecologist (who does not work on this topic) want to cite it in a general way. Maybe asking for comments from colleagues who work on other topics will help.

I suggest revision of the manuscript and have provided some more specific comments below. This is an impressive study and certainly interesting, but just feels to me as though that is not being sufficiently communicated!

We appreciate the positive comments from the reviewer. We have addressed all concerns of the reviewer below. While we did add additional examples, e.g., from a tropical freshwater ecosystem (Lines 44-47), to the introduction, we did not overhaul the writing as to not substantially change the manuscript in a way that would compromise the aspects that other reviewers have pointed out are strengths.

L 34: Do these results really suggest that “phylogenetic conservatism is a critical determinant”, or does it suggest that phylogenetic position is a better predictor than ecological factors regarding nutrient dynamics on coral reefs? To me, it seems the latter.

We agree with the reviewer and also like their suggested phrasing. We have changed this to “phylogeny is a critical determinant of...” because indeed it is a better predictor than ecological factors, but it is also important to note that it is in fact a good predictor in of itself. We could not add further changes due to word limitation’s in the abstract. Line 35

L 43: Maybe this is a bit cosmetic, but I think for people outside this specific field, an actual example would also be helpful here (rather than a description of a function labelled as an example).

We agree fully and have added a specific example from tropical freshwater ecosystem in attempt to broaden the scope of the MS. Lines 44-47

L 49: Are nitrogen, phosphorous and carbon “chemical traits”? That seems like an odd description to me.

We agree with the reviewer and struggled with this term ourselves, largely because we do not want to introduce more jargon into the literature. However to be clear, we do not claim that nitrogen, phosphorus, and carbon themselves are chemical traits, but instead the storage of these elements in an organisms tissue, or their supply (we posit) is. For this reason, we have continued the use of this term, but are happy to reconsider if the editor believes this change is necessary.

L 56-57: This example (and especially that from 59-62) is better than that on 43, as it allows a non-specialist in this particular field to appreciate exactly what’s going on.

Thank you, we hope that the new example we used in line 44 is also helpful.

L 108: Why wasn’t “geographic location” (Moorea vs Caribbean) included as a fixed effect? It would also be informative to have an idea of the phylogenic and “ecological” diversity of the species sampled from the Carribbean and Moorea relative to the species that exist in those regions – i.e. how skewed were the data? I understand this was addressed in latter parts of the study, but why was it excluded here as it might help explain the results?

Geographic location was included as a fixed effect in the model (it is designated by the term “region” in Figure 1). In previous iterations of the manuscript we included a paragraph at the beginning of the results section that outlines the entire dataset – including information about the number of species to which the reviewer is referring. Given the journal format however, we moved this text to the beginning of the results section. As suggested by the reviewer, we have added a clear statement that “region was included as a fixed effect” (Line 134), and we have added the number of species found in each region as well as the number of species we collected in the Methods section. Lines 374-375.

L 165: Might be a dim question, but how is there a range for 62-77% - shouldn't that be one number?

We agree, this was confusing, and we have clarified this text. Line 212.

L 185 and L 190 both start with “collectively”

Thank you for noticing this. We have omitted the second “collectively”.

L 197: I would suggest you remove the word “exceptional” – too emotive

We agree that this is unnecessary. We have omitted this word.

Reviewer #3 (Remarks to the Author):

Allegeir et al. set out to explore the factors that determine organismal chemical traits across two distinct geographic regions. Specifically, they explore if phylogenetic relatedness or various ecological factors drive fish chemical traits. The authors do a nice job bringing in both evolutionary and ecological literature and work to bridge these fields in this study. In comparison to some of these authors' previous work (Allgeier et al. 2020), I find these analyses to be more robust and convincing. I thought this was a strong study and was strengthened by the use of communities from two distinct regions with unique geologies and background nutrient limitation status.

We thank you for the positive comments about our manuscript.

The article provides novel results and the conclusions are backed up by the data. While I thought the analyses were appropriate, I think this paper would better highlight the phylogenetic conservatism suggested here by providing a phylogenetic tree of the study species from the two regions and somehow highlight or map the traits onto this tree to better visualize how these traits are conserved across the evolutionary lineages. This may not need to be in the main text, but included as a supplement. However, I think this would better highlight the relatedness amongst these families and how the traits are actually conserved across time. With that, is there a dated phylogeny of these species as suggested in the text regarding the time since divergence?

We thank you again for your positive comments. We also agree with the reviewer that visualizations of the data would be a nice addition to the manuscript. To that end, we now include phylogenies for each region, with each trait mapped onto the phylogenies as

supplemental figures. We have also expanded and clarified the description of the phylogeny used in the main text, now explicitly stating that the phylogeny is dated, and providing the reference clarifying that there is a published dated timetree that we are using. Lines 446-451

Some more specific comments are highlighted below.

In regards to the methods, did the authors note any inter-annual variation in the traits they measured? For example, different chemical analyses were used in different years.

All chemical analyses were conducted in all years, that is, measurements for all traits were conducted in all years. The primary factor that may have differed among years could have been subtle variation in temperature for the excretion experiments. However, we took great care to control for temperature in all our experiments and in all cases were consistent across time within each region. Additional factors that could have potential influence on chemical traits would be if some event (e.g., large-scale algal bloom) cause some type of shift in feeding behavior of the fishes. However, our analysis, which is consistent with previous studies (Allgeier et al. 2015, 2020, Vanni and McIntyre 2016) shows that changes in diet, and for that matter, changes in the ecology, should not have major impacts on chemical traits. Nonetheless, to the best of our knowledge there were no such events between the years in which we sampled in the different regions. Further, previous analyses of the Caribbean does not indicate any significant effect of the date of collection on excretion rates or body chemistry of species for which we have a large number of samples (Allgeier unpublished).

Allgeier, J. E., S. J. Wenger, D. E. Schindler, A. D. Rosemond, and C. A. Layman. 2015.

Metabolic theory and taxonomic identity predict nutrient cycling in a diverse food web. Proceedings of the National Academy of Science 112:2640–2647.

Allgeier, J. E., S. Wenger, and C. A. Layman. 2020. Taxonomic identity best explains variation in body nutrient stoichiometry in a diverse marine animal community. Scientific Reports 10:13718.

Vanni, M. J., and P. B. McIntyre. 2016. Predicting nutrient excretion of aquatic animals with metabolic ecology and ecological stoichiometry: a global synthesis. Ecology 97:3460–3471.

Line 360: Can more details be provided regarding the phylogenetic covariance matrix? For example, are these maximum likelihood or Bayesian covariance matrices?

We agree that this is an important point to clarify and this was also pointed out by Reviewer 1. We used the phylogeny that only included species with genetic data, not the phylogeny that included taxa placed using stochastic polytomy resolution. We have clarified this in the manuscript by expanding the description of the phylogeny and referencing the paper that published the phylogeny using species with genetic data only, which we had previously omitted. Lines 446-452

Line 384: Can the authors be more clear how isotope values were standardized? Are they simply stating since they did not examine the centroid and only the distribution of data (i.e., amount of area or variance) that is how it was standardized? This could be more clear.

We apologize for lack of clarity. Isotopic data was standardized using a z-score approach ($[x - \text{mean}(x)] / \text{sd}(x)$) for each region in all cases. Doing so provides a relative comparison of the spread of the isotopic data for each region. Even after doing this we did not make any absolute comparisons across the regions as this could be confounded by varying isotopic baselines in each region. We have added more text for clarity. Line 475

In figure S2, I found it curious that the communities with the higher K-statistic and Pagel's values were less-likely (as indicated by the proportions above) to be different from a randomly simulated community. What is driving that? Also, the values at the top appear to be percentages as opposed to proportions.

The reviewer raises an interesting question and, while we do certainly see the Reviewer's point, we also note that this observation is not always true, e.g., in all but three cases the full dataset, which always has a substantially larger K or Pagel's value than the Polynesia data, has an equal or higher chance of differing from a randomly simulated community than the Polynesia data. As for the cases in which higher K or Pagel's value are associated with being less likely to be different from the randomly simulated community, we generally interpret this as a function of larger variation in the Caribbean data than the Polynesian data. Each statistic (K-statistic and Pagel's) was calculated using a mean value for each species. However, because we had multiple individuals per species in our dataset, we wanted to make sure we were accounting for potential variation among individuals that may affect the species-level means. Thus this iterative process does indeed reveal that the Caribbean data tends to have greater variation (also noted by the larger error bars) that results, at times, in the outcomes not differing from randomly simulated communities.

What do the authors mean by phylogenetic history on line 27? Phylogenetic relatedness would be more accurate.

We agree and have changed "phylogenetic history" to "phylogenetic relatedness".

Lines 90-91: Are there citations for these statements regarding nutrient limitation differences in the two study regions?

Thank you for picking up on this. We have added the appropriate citations here.

Line 201: What chemical traits? Be specific. It would be helpful to define which ones varied.

We agree and have added which specific chemical traits to the text. Line 254

REVIEWERS' COMMENTS

Reviewer #1 (Remarks to the Author):

General comments

I reviewed the manuscript by Allgeier et al. in the first round (Reviewer #1) and am glad to see that most of my initial concerns have been addressed and/or clarified by the authors. I have carefully read their responses to all of the reviewers and, in general, they provided detailed comments and made some relevant changes that helped improving the work. Despite this, I still have some minor points (mostly regarding text changes) that require attention by the authors. I am happy to recommend this work for publication after these minor comments have been addressed.

Minor Points

In the abstract you jump from talking about the importance of nutrient cycling for reefs (Lines 21-22) to what you did (Lines 22-27). This feels a bit abrupt, so maybe just add the word 'Here' in your second sentence to help with the flow. The same applies for the last paragraph of the introduction (Lines 84-89). But in this case, maybe just separating the paragraph in two would help: one about nutrient cycling on coral reefs and the last one with your study objectives.

Still in the last paragraph of the introduction (Lines 94-96), this sentence is technically incorrect. Your study included fishes that have actually very distinct biogeographic histories, but have similar environmental and ecological demands. The way it is written can be misleading, so I suggest rephrasing: 'Importantly, our study included fishes that have similar environmental and ecological demands (i.e., temperature, habitat types), but contrast in...'.

The first paragraph of the results is a succinct description of the methods, so it might be unnecessary because it repeats what is better described in the appropriate section.

In lines 160; 186; 192 you use the expression 'phylogenetically-corrected' to refer to the models, but I think a more appropriate expression would be 'phylogenetically-informed'.

Please, update the number of families throughout the manuscript. Although you have updated the number when you mention the shared families (from 12 to 11), the total number of families should also be updated (from 41 to 40) after dropping 'Scaridae'. The familial status of species should also be updated in your Table in the Supp material and in the legend of figure 3.

Finally, it is important to specify how you assessed convergence in your Bayesian models in the methods section. This is quite a crucial information for reproducing your model diagnostics, and it is currently missing.

Line 1: Should the title be 'in coral reef fishes'?

Lines 24; 331; 332: Change 'fish' to 'fishes'. Fish can be used as a plural when you are talking about individuals within the same species, but in those cases you refer to individuals in different species so the correct use is 'fishes'.

Line 27: I think 'on coral reefs' can be deleted from this sentence to make it shorter.

Line 67-69: Suggested rephrasing to avoid the repetition of species and shared: 'Conversely, as taxonomic groups (e.g. families) accumulate species, the shared evolutionary and biogeographic histories of lineages can result in traits being similar within a clade^{12,13}'.

Line 124: Please, define the acronym in the first use: deviance information criterion (DIC).

Line 131: 'results suggest' instead of 'indicate', to follow my comment on the first round that high proportion of variance explained by phylogenetic relatedness in hierarchical models does not necessarily equate to phylogenetic signal.

Line 147: parenthesis missing.

Line 164: extra dot.

Line 213: '...within shared families.'

Line 222: 'However' to avoid repeating 'Yet', and I would remove the 'however' from line 224.

Line 244: replace 'fish community' for 'assemblage'.

Line 268: C:P and N:P.

Line 283: 'considered in light of our other findings here' can be removed to reduce the sentence.

Line 310: '(see Methods and SI for further detail)', is there any method description in SI? Also 'see Methods' in the Methods section is unnecessary.

Line 316: '805 individuals in 107'

Line 365: As stressed in my comment on the first round, the 'fishtree' R package does not generate covariance matrices, so please remove this sentence.

Lines 367-371: Suggested rephrasing: 'The phylogeny was made ultrametric using non-negative least squares⁵⁶, and then converted into an inverted phylogenetic covariance matrix using the algorithm from ⁵⁷ implemented with the "inverseA" function from the MCMCglmm package in R⁵⁸. This covariance matrix was then incorporated as a random effect in Bayesian phylogenetic mixed model.'

Legend Figure 1: 'Phylogenetic relatedness' instead of 'Phylogenetic conservatism'

Reviewer #2 (Remarks to the Author):

Review 2 of: Phylogenetic conservatism drives nutrient dynamics of coral reef fishes

This is the second time that I have reviewed this paper. I think that the authors did a fantastic job addressing my (Reviewer 2) comments, and they also appear to have adequately addressed the concerns raised by the other reviewers. I have a few minor comments, which I have detailed below. Generally, I do think that another good edit of the text would be helpful as it still is a bit "clunky" in parts. Outside of this, I think the authors have done a good job and I am happy to recommend acceptance.

L 65: I'd suggest changing the word "significant" here.

L 80: A statement such as this probably warrants a citation.

L 82: I'd change "ecosystem" to "area" at the end of this sentence.

L 83-84: Maybe, "within the ecosystem's biomass" rather than "within ecosystem biomass"

L 85: I'd suggest changing that dash to a comma and removing the comma following tissue

L 95: You seem to switch between including i.e. in and out of brackets. Personally I'd choose one way and stick with it.

L 100: The "etc" makes it seem a little too casual to me.

Reviewer #3 (Remarks to the Author):

I appreciate the care that the authors took in addressing my comments and the other reviewers particularly in regards to the phylogenetic matrix. I do not have any major reservations regarding this manuscript. I feel as though they adequately tested their questions and hypotheses. Additionally, I appreciate the additional figures highlighting the traits map on the phylogeny. I found the conclusions backed up strongly by their data.

REVIEWERS' COMMENTS

Reviewer #1 (Remarks to the Author):

General comments

I reviewed the manuscript by Allgeier et al. in the first round (Reviewer #1) and am glad to see that most of my initial concerns have been addressed and/or clarified by the authors. I have carefully read their responses to all of the reviewers and, in general, they provided detailed comments and made some relevant changes that helped improving the work. Despite this, I still have some minor points (mostly regarding text changes) that require attention by the authors. I am happy to recommend this work for publication after these minor comments have been addressed.

Thank you for your positive comments.

Minor Points

In the abstract you jump from talking about the importance of nutrient cycling for reefs (Lines 21-22) to what you did (Lines 22-27). This feels a bit abrupt, so maybe just add the word 'Here' in your second sentence to help with the flow. The same applies for the last paragraph of the introduction (Lines 84-89). But in this case, maybe just separating the paragraph in two would help: one about nutrient cycling on coral reefs and the last one with your study objectives.

This comment is no longer relevant as the abstract was shortened per journal requirements.

Still in the last paragraph of the introduction (Lines 94-96), this sentence is technically incorrect. Your study included fishes that have actually very distinct biogeographic histories, but have similar environmental and ecological demands. The way it is written can be misleading, so I suggest rephrasing: 'Importantly, our study included fishes that have similar environmental and ecological demands (i.e., temperature, habitat types), but contrast in...'

We agree and have adjusted the wording here (line 93)

The first paragraph of the results is a succinct description of the methods, so it might be unnecessary because it repeats what is better described in the appropriate section.

This paragraph was not altered.

In lines 160; 186; 192 you use the expression 'phylogenetically-corrected' to refer to the models, but I think a more appropriate expression would be 'phylogenetically-informed'.

We have kept the wording as is.

Please, update the number of families throughout the manuscript. Although you have updated the number when you mention the shared families (from 12 to 11), the total number of families should also be updated (from 41 to 40) after dropping 'Scaridae'. The familial status of species should also be updated in your Table in the Supp material and in the legend of figure 3.

Thank you for catching this. This was corrected throughout.

Finally, it is important to specify how you assessed convergence in your Bayesian models in the methods section. This is quite a crucial information for reproducing your model diagnostics, and it is currently missing.

We have added text to clarify this. Lines 371, 428-9.

Line 1: Should the title be 'in coral reef fishes'?

This was not changed

Lines 24; 331; 332: Change 'fish' to 'fishes'. Fish can be used as a plural when you are talking about individuals within the same species, but in those cases you refer to individuals in different species so the correct use is 'fishes'.

This was changed.

Line 27: I think 'on coral reefs' can be deleted from this sentence to make it shorter.

This was not removed.

Line 67-69: Suggested rephrasing to avoid the repetition of species and shared: 'Conversely, as taxonomic groups (e.g. families) accumulate species, the shared evolutionary and biogeographic histories of lineages can result in traits being similar within a clade^{12,13}'.

We agree. Aspects of this text was modified.

Line 124: Please, define the acronym in the first use: deviance information criterion (DIC).

This text was added.

Line 131: 'results suggest' instead of 'indicate', to follow my comment on the first round that high proportion of variance explained by phylogenetic relatedness in hierarchical models does not necessarily equate to phylogenetic signal.

This text was changed

Line 147: parenthesis missing.

Added

Line 164: extra dot.

Removed

Line 213: '...within shared families.'

This was added

Line 222: 'However' to avoid repeating 'Yet', and I would remove the 'however' from line 224.

We agree and have adjusted the text accordingly.

Line 244: replace 'fish community' for 'assemblage'.

This was changed

Line 268: C:P and N:P.

This was changed.

Line 283: ‘considered in light of our other findings here’ can be removed to reduce the sentence.
This was removed.

Line 310: ‘(see Methods and SI for further detail)’, is there any method description in SI? Also ‘see Methods’ in the Methods section is unnecessary.
This was changed.

Line 316: ‘805 individuals in 107’
This text has been reworded correctly.

Line 365: As stressed in my comment on the first round, the ‘fishtree’ R package does not generate covariance matrices, so please remove this sentence.
This was omitted.

Lines 367-371: Suggested rephrasing: ‘The phylogeny was made ultrametric using non-negative least squares⁵⁶, and then converted into an inverted phylogenetic covariance matrix using the algorithm from ⁵⁷ implemented with the “inverseA” function from the MCMCglmm package in R58. This covariance matrix was then incorporated as a random effect in Bayesian phylogenetic mixed model.’
We have accepted this suggested wording.

Legend Figure 1: ‘Phylogenetic relatedness’ instead of ‘Phylogenetic conservatism’
This was changed.

Reviewer #2 (Remarks to the Author):

Review 2 of: Phylogenetic conservatism drives nutrient dynamics of coral reef fishes

This is the second time that I have reviewed this paper. I think that the authors did a fantastic job addressing my (Reviewer 2) comments, and they also appear to have adequately addressed the concerns raised by the other reviewers. I have a few minor comments, which I have detailed below. Generally, I do think that another good edit of the text would be helpful as it still is a bit “clunky” in parts. Outside of this, I think the authors have done a good job and I am happy to recommend acceptance.

We thank the reviewer for their positive comments and their useful previous reviewer suggestions.

L 65: I’d suggest changing the word “significant” here.
This was changed to ‘relevant’.

L 80: A statement such as this probably warrants a citation.
A citation was added.

L 82: I’d change “ecosystem” to “area” at the end of this sentence.
This was changed to “environments”.

L 83-84: Maybe, “within the ecosystem’s biomass” rather than “within ecosystem biomass”
This was not changed.

L 85: I’d suggest changing that dash to a comma and removing the comma following tissue
This was changed.

L 95: You seem to switch between including i.e. in and out of brackets. Personally I’d choose one way and stick with it.
Thank you, we agree and have done so.

L 100: The “etc” makes it seem a little too casual to me.
This was not changed.

Reviewer #3 (Remarks to the Author):

I appreciate the care that the authors took in addressing my comments and the other reviewers particularly in regards to the phylogenetic matrix. I do not have any major reservations regarding this manuscript. I feel as though the adequately tested their questions and hypotheses.

Additionally, I appreciate the additional figures highlighting the traits map on the phylogeny. I found the conclusions backed up strongly by their data.

We thank the reviewer for their positive comments and their useful previous reviewer suggestions.